# Improving Robot Predictability via Trajectory Optimization Using a Virtual Reality Testbed

Clare Lohrmann
clare.lohrmann@colorado.edu
CU Boulder
Boulder, Colorado, USA

Ethan Berg
ethan.berg@colorado.edu
Fairview High School
Boulder, Colorado, USA

Bradley Hayes
bradley.hayes@colorado.edu
CU Boulder
Boulder, Colorado, USA

Alessandro Roncone
alessandro.roncone@colorado.edu
CU Boulder
Boulder, Colorado, USA

## ABSTRACT

The ability to predict where a robot will be next, or how it will navigate an area is critical to safe and effective human-robot collaboration and interaction. Due to information asymmetry, the path that a robot takes may be optimal, yet unpredictable to an observing human who does not have access to the same information. Unpredictability presents a safety risk to humans, and also makes interacting with robots more cognitively intensive and confusing than need be. In this work, we propose an algorithm that optimizes a robot's trajectory for predictable behavior, resulting in a robot that moves in a way that is more predictable to humans, balanced with what is optimal to the robot. To validate this approach, we propose two human-subjects experiments, one of which is conducted in virtual reality.

## CCS CONCEPTS

• **Human-centered computing** → **Virtual reality**; • **Computing methodologies** → **Robotic planning**; **Optimization algorithms**.

## KEYWORDS

Predictability, Virtual Reality, Trajectory Optimization

**ACM Reference Format:**
Clare Lohrmann, Ethan Berg, Bradley Hayes, and Alessandro Roncone. 2024. Improving Robot Predictability via Trajectory Optimization Using a Virtual Reality Testbed. In *Proceedings of 7th International Workshop on Virtual, Augmented, and Mixed-Reality for Human-Robot Interactions (VAM-HRI '24).* ACM, New York, NY, USA, 6 pages. https://doi.org/XXXXXXX.XXXXXXX

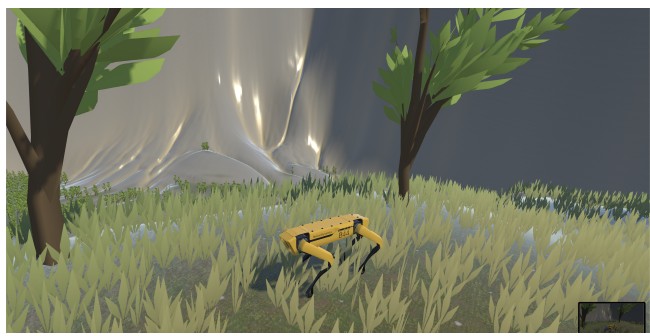

**Figure 1: A to-scale Spot robot as viewed via the Meta Quest 3. The robot traverses the simulated environment, enabling testing of our algorithm for robot trajectory optimization in a robust and realistic environment.**

## 1 INTRODUCTION

As robots become more integrated into additional domains and aspects of daily life, the problem of robot unpredictability has become more salient. Prior work has indicated that predictability of

robots is critical to smooth and effective human-robot collaboration and interaction. Humans find predictable robots easier and more satisfying to work with - they trust them more and view them more positively than less predictable robots [5].

Predictability, the "quality of matching expectation" [6], is the key to successful human-robot interactions, as research has shown that human expectations of robots are usually different from the robot's capabilities. Calibrating and reducing the gap between expectations and reality is the challenge that this work addresses via predictability. By emphasizing predictability with minimal sacrifice of optimality, we can more effectively collaborate and match human expectations with robot performance.

There is a significant body of work that explicitly changes robot behavior to follow a convention or heuristic to promote human comfort or smoother collaboration [4, 17, 18], but these methods are pre-programmed and are not flexible to new environments. Other work has been done with predictability on the task-level, as well as the motion planning-level [6, 7].

There have also been prior works that use reinforcement learning approaches for human-robot or human-agent collaboration. However, many of these approaches are not validated with human subjects nor are they flexible - the agent's policy is learned, and cannot be easily adjusted to new scenarios where the environment or goals have changed [9, 11, 19]. Few approaches are validated

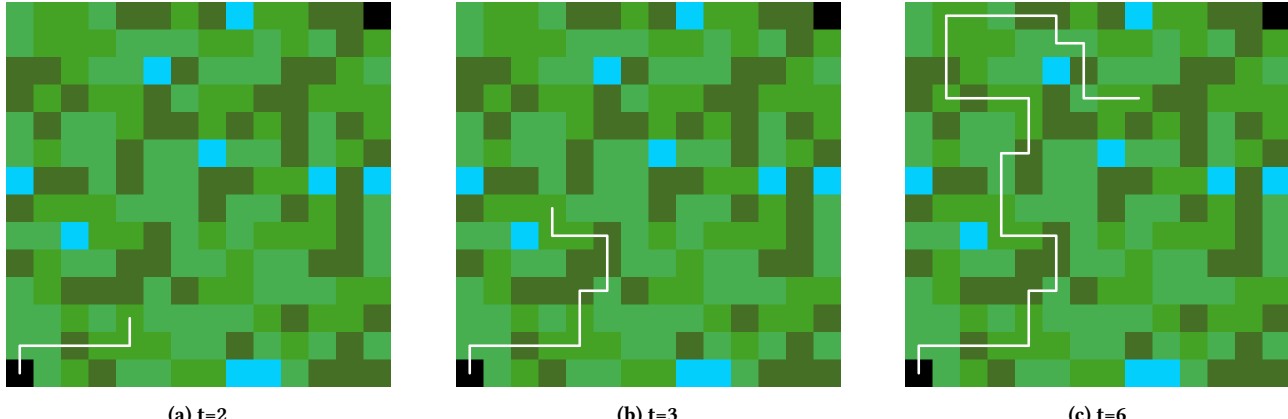

(a) t=2                          (b) t=3                          (c) t=6

Figure 2: These figures illustrate the optimal trajectory of a robot through a given environment. While the robot is reasoning over all information available to it, this information may not be visible to the human, as in this case. While the human may be able to see obstacles (blue), areas of small negative reward that the robot may be avoiding could appear as normal areas (shades of green). Because of this information asymmetry the trajectory the robot takes does not match expectations.

across multiple environments, in the context of a human subjects study, with an embodied or simulated robot, as this work does.

The predictability function we introduce emphasizes the symmetry of the generated path, such that it better matches human expectations, given humans' preferences for symmetry [21, 23]. We formulate our approach as a multi-objective trajectory optimization problem. This allows us to create objectives for more practical aspects, such as collision avoidance and state validity, as well as balancing predictability and optimality simultaneously within the same interaction.

Through the use of VR, we validate our algorithm while simultaneously ensuring participant safety, which would be challenging to do in a real environment with a physical robot. Additionally, we are able to construct a larger variety of environments for validation, in challenging environments such as those with obstacles and situations that are difficult for physical robots to traverse. Through this validation, we expect to show that our approach facilitates more predictable, understandable, and effective human-robot collaboration, while retaining robot performance, balancing predictability and optimality.

## 2 RELATED WORK

A significant problem in facilitating effective human-robot collaboration is the incorporation of mental models. Mental models are structures about the world that humans construct in their minds to help navigate environments, make decisions, and reason about collaborators [20]. Humans are exceedingly skillful at constructing mental models about other human collaborators [24]. Human factors research indicates that the more accurate a human's mental model of another is, the better they will be able to collaborate with each other [14]. This concept does translate into human-robot interactions, as humans also construct mental models of the robots that they work with [20]. As in human-human collaboration, the more accurate the human's mental model of the robot is, the more

effective the human-robot collaboration will be [10, 14, 16, 25]. Going further, when humans and robots form shared mental models, the collaboration is even better [16].

We adopt Dragan's definition of predictability as "the quality of matching expectations" [6], which is distinct from a concept such as legibility. For the purposes of this work, we operate under the realistic assumption that human collaborators are aware of robot teammates' goals. Prior work underscores the importance of predictability in human-robot interaction, as robots matching the expectations of humans leads to better collaboration [10, 14, 16, 25]. There are also significant downstream implications of predictability. Agents and robots that are more predictable are preferred, and more trusted by the humans that interact with them [5].

The expectations that a human has of a given robot are derived from their mental model of the robot [20]. Thus, when the robot is more predictable to the human, this indicates that their mental model of the robot is more accurate, which leads to the positive consequences discussed earlier. Some prior work has shown that emphasizing predictability above or in tandem with optimality leads to better human-robot collaboration [7], but this domain is relatively underexplored.

There is a significant obstacle to humans forming accurate mental models of robots. Humans and robots reason in fundamentally different ways. Robot programming usually relies on algorithmic and mathematical approaches. Humans, by contrast, do not optimize [8]. Rather, they rely on a cognitive toolkit that contains strategies such as heuristics and patterns [1, 2, 8, 13, 15, 22].

Ergo, the human and robot are reasoning in different ways about a shared environment, which leads to a significant mismatch between the human's model of the robot and the ground truth. There have been works that focus on altering the human's perceptions of the robot [3, 12], but human cognitive methods, refined over millennia of evolution, are far less flexible than altering the robot's behavior to match human expectations.

In order to emphasize the predictability of the robot while also retaining strong robot performance, we formulate our solution as a multi-objective optimization problem, which takes the optimal path as an input. Formulating our solution in this way allows for greater flexibility than other methods, such as reinforcement learning, not requiring substantial offline computation to accommodate changes and not being restricted by the Markov property. In this formulation, the cost of a given waypoint can be derived using any number of other waypoints and the state size can be variable depending on both the objective as well as the environment. The size of the environment can also be highly variable without making any adjustment to the objective function or any retraining. Additionally, our formulation is independent of the method used to obtain or encode an optimal robot policy. Our approach doesn't rely on knowledge of the optimal policy itself, only its output (a rollout of the robot's trajectory). Lastly, our proposed formulation for navigation problems utilizes a simplified environmental representation such that there are two types of states within the environment: impassable terrain states (obstacles) and normal states. Additional state types may be added as applications require, but this is the minimal set required.

To validate our approach, we use virtual reality to simulate a variety of outdoor environments. Each environment a participant sees is procedurally generated such that while participants may see environments that follow the same template (i.e. placement of obstacles and areas of negative reward to be avoided by the optimal robot), no two environments are identical, removing concerns of using cherry-picked scenarios.

Additionally, the virtual environments allow for a greater variety of obstacles and terrains than would be possible to run outside of virtual reality. Participants can navigate around lakes, over boulders, through dense brush, as well as through steep and uneven terrain without the risk of personal injury or damage to the robot.

## 3 METHODS

We define a trajectory as an ordered list of waypoints. The waypoints are connected by segments, with timing information informing the duration of transition.

The cost of a given trajectory is the sum of the cost of each of the waypoints. Our method minimizes the cost using a stochastic trajectory optimization process.

The cost of a given waypoint is formulated as a summation of individual objective functions. Some of the functions are considered to be harder constraints than others. Several of the objective functions involve obeying the basic mechanics of the environment, such as collision avoidance and restricting the robot to valid states. These functions ensure that the final trajectory obeys the mechanics of the environment and limitations of the agent. The other components of the objective function relate to the balance of predictability and optimality of the trajectory. The optimality component relies on the output of the optimal policy, and prevents the predictable trajectory from straying too far from the optimal policy. The predictability component minimizes cost for those waypoints that promote symmetry in the trajectory. This leads to trajectories that are more symmetrical, and thus more visually appealing and predictable to humans.

### 3.1 Collision Avoidance Objective

$$C(w) = C_{waypoint}(w) + C_{segment}(w)$$

$$C_{waypoint}(w) = \begin{cases} 1 & w \text{ is within or on an obstacle} \\ 0 & w \text{ is within an open space} \end{cases}$$

$$C_{segment}(w) = S_{num}(w)$$

The collision avoidance objective is formulated as a sum of two components - the waypoint collisions and the segment collisions. The waypoint collision cost of a given waypoint is relatively simple - if the waypoint is within or touching an obstacle/impassable state, the cost is 1. Otherwise the waypoint collision cost is 0. The segment collision cost is more complex and can be omitted depending on the methodology used to traverse between waypoints. If a simple interpolated path is used to connect waypoints, the segment cost should be used. This cost assesses the segments connecting the given waypoint with the prior and subsequent waypoints. The segment collision cost for a given waypoint is the number of obstacles crossed by either adjoining segment, indicated by $S_{num}$. This cost may be omitted if the connections between waypoints are derived via a more expressive method (e.g., RRT* motion plan).

### 3.2 Valid State Objective

This objective assesses a penalty to any waypoint that is placed outside the bounds of the given environment. The valid state cost for a given waypoint is 0 if the waypoint is within the environment, and 1 if it is outside the allowable values.

### 3.3 Connectivity Objective

We want the waypoints to be accessible via the movements the agent can make, so the waypoints must be within a certain distance of each other. In the case of the environments used in our testbed, this maximum distance is unit length. The cost is therefore 0 if the waypoint is not more than the maximum distance from the prior and subsequent waypoints. If the waypoint is more than the maximum distance from the prior or subsequent waypoints, the cost of the given waypoint is the distance overage. This prevents the final trajectory from being unachievable by the agent.

### 3.4 Distance from Optimal Objective

The distance from the optimal trajectory is used to balance between optimality and predictability. By minimizing the distance between a trajectory maximizing predictability and the original optimal trajectory, the final trajectory will not sacrifice all performant behavior for predictability. As the predictable trajectory has the same number of waypoints as the optimal path, this cost is the sum of the distance between corresponding waypoints in the predictable trajectory and the given optimal one. This objective function allows us to keep optimality as a factor in the trajectory, thus sacrificing less reward when creating a more predictable path. This component may also be weighted as 0 if a purely predictable path that does not consider optimality is desired.

### 3.5 Symmetry Objective

Our main method of promoting predictability within the trajectory is promoting symmetry. We do this by determining the longest repeating sequence of actions. In our use case, the waypoints of

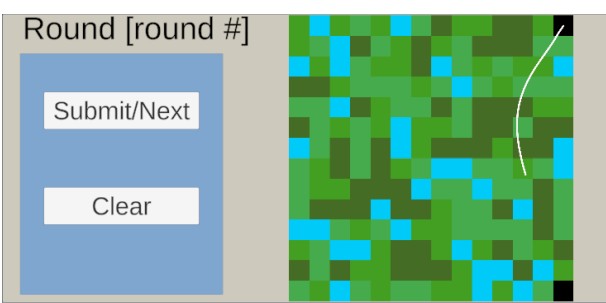

**Figure 4: A visual illustrating the interface to conduct the online study, also developed via simulation. Participants will complete the robot's trajectory to the goal in the way they believe the robot will move, and will be assigned a score based on their accuracy at predicting how the robot will get to the goal.**

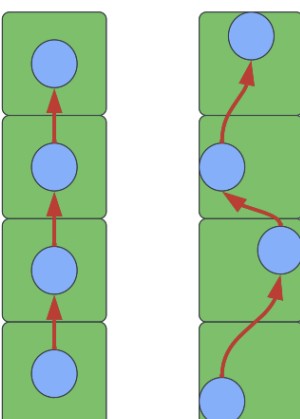

**Figure 3: In this scenario, if we simply used the actions connecting states for the symmetry cost rather than directional change, both of these trajectories would be the same sequence of actions, and the same cost, even though one has more symmetry than the other.**

the trajectory should be at most one action apart from the previous waypoint, due to the limitations of our agent. Thus, the trajectory can be converted to a sequence of actions relatively easily. When waypoints are moved during optimization, they may have a corresponding action that connects them, but this cannot be guaranteed. The waypoints may be incurring a connectivity cost, and be unconnectable via an action. Additionally, as waypoints are moved in opposite directions, they may still be connectable via the same action, though the trajectory has changed, as in Figure 3. Therefore, we use a more fine-grained action space, such that the agent's actions are broken down into multiple actions related to the angle of change between waypoints (directional change). This allows for only the most symmetrical trajectories to incur no cost.

The sequence of actions is run through a longest repeating substring algorithm to determine the longest sequence of actions that is repeated during the trajectory. Then, for each waypoint, if the waypoint is part of this subsequence, the cost incurred by the waypoint is 0, and is 1 if the waypoint is not part of the subsequence. We also evaluated using the most repeated substring, which does create some improvement in the predictability of the trajectory, but was not as effective as the longest repeated substring across the wide diversity of generated environments tested.

## 4 EVALUATION

We plan to conduct two user studies, one online and one in-person using virtual reality. Each study is designed to test different effects of the use of our algorithm. The online study explicitly tasks participants with predicting the robot's route to the goal multiple times at various timesteps for each environment. Participants in this study will also be surveyed about predictability, understandability, and teaming metrics, but the lack of embodiment places the emphasis on our algorithm's predictability. The second study will be conducted in-person using VR to simulate an outdoor environment with a to-scale Boston Dynamics Spot robot. In this scenario, participants will not be able to visualize the robot's trajectory and instead will be tasked with implicitly predicting the robot's trajectory by staying within a certain radius of it while completing their task. VR participants will also be asked about the robot's predictability and understanding of the robot's behavior, as well as the performance of the team. We will collect objective metrics of participants' ability to predict the robot's movements, but this experiment emphasizes the human's experiences when working with a robot implementing our algorithm.

## 5 EXPERIMENTAL DESIGN
### 5.1 Game Environment

For the VR experiment, we construct an interactive simulation, deployed on the Meta Quest 3. In the simulation, participants and the robot will traverse the environment to a goal point that is known to both. While the goal point is known to the participant, the path the robot takes to get there will be unknown. As the robot and participant make their way through the environment, the participant will be tasked with short retrieval errands. While the participant retrieves items in the environment (in this case rock samples), the robot will continue along its path, requiring participants to effectively reason about the robot's trajectory in order to rendezvous back with the robot. Areas of negative reward for the robot's policy, such as steep slopes and uneven terrain, are visible to the user though the robot's policy is not explicitly explained. Robots following the optimal policy will avoid such areas wherever possible, our algorithm does not. Participants will be scored based on how quickly they are able to return rocks to the robot, as well as how close they remain to the robot throughout the experiment. Participants will engage in six rounds of approximately four minutes of gameplay each within environments that will be procedurally generated at runtime.

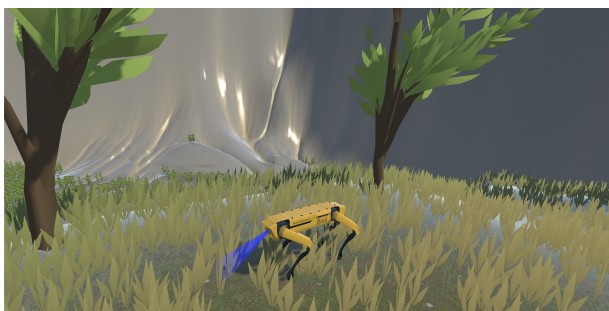

**Figure 5: The to-scale Spot robot scanning for rocks in the simulated environment. By engaging with the robot in VR, participants can engage safely with the robot while still indoors.**

## 5.2 Experimental Design

The experiment will be conducted with each participant randomly assigned to one of two groups. Optimal: in this condition, the Spot robot will follow the optimal trajectory through the environment, avoiding all areas of negative reward. Predictable: in this condition the Spot robot will follow the trajectory generated by our algorithm, which while suboptimal, is designed to be more predictable to the human teammate.

## 5.3 Hypotheses

We will conduct an IRB approved human-subjects study to investigate the following hypotheses regarding the effectiveness of our method within a human-robot collaborative game:

- $H_1$: Participants in the predictable condition will achieve higher scores in the game environment, as they will be more efficient in rendezvousing with the robot.
- $H_2$: Participants in the predictable condition will complete the task with lower scores for cognitive fatigue.
- $H_3$: Participants in the predictable condition will view the robot more positively (predictable, understandable, intelligent, etc).

Through the use of this virtual environment, we will be able to validate the algorithm in a setting that would not otherwise be possible. Participants will engage with a to-scale robot in an outdoor setting, rife with obstacles, visual obstructions, and varying terrains. No two participants will see the same environment as they are procedurally generated. This will allow for a more robust test bed, and further illustrate the applicability of this algorithm in real-life situations.

## 6 CONCLUSIONS

In this work, we proposed an approach to improve the predictability of robot teammates during human-robot collaboration and described an experimental setup that uses virtual reality to simulate more realistic and unconstrained collaborative environments for testing. We hypothesize that participants engaging with the robot in these virtual environments will find the robot using our method more predictable and easier to work with. The use of virtual reality

will allow participants to engage with the robot safely and effectively in a wide variety of environments and terrains, which would not be feasible without VR.

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
