# OpenReview forum: "Improving Robot Predictability via Trajectory Optimization Using a Virtual Reality Testbed"
_humanrobotinteraction.org/HRI/2024/Workshop/VAM-HRI — VAM-HRI 2024 Oral_

### Official Review · Reviewer_DpSN · 2024-02-23
**Accept**

**Rating:** 8
**Confidence:** 4

**Review:**

This paper proposes a new algorithm that optimizes a robot’s trajectory for predictable behavior by using symmetry to appeal to users. To evaluate the effectiveness of this algorithm, an online user study and an in-person user study are proposed, and will compare an optimal path versus their more symmetrical generated path. In the second study, participants will be completing a task in simulation within a VR environment with the robot.

Strengths:

- This paper provides a highly interesting way for researchers to evaluate their social robot navigation algorithms in VR. What may be particularly impactful are the steps for procedurally generating new environments. I would be eager to see how I might incorporate this environment into my own work and hope the authors provide an open source implementation.

- This work provides an interesting step towards making robots predictable through symmetry. I personally have not seen much work with this approach.

Areas of Improvement:

- In Figure 2, it would be helpful if the symmetric paths were overlaid on top of the optimal path. Currently, it is unclear how the symmetric path would differ from the optimal one. For example, if the robot believes it is walking in the wrong direction, will it keep walking forward for the sake of symmetry? If it turned out it did walk the wrong direction and needed to backtrack, I would assume this would make the robot less predictable and trustworthy to the users.

- In the visual for Figure 4, it is unclear what the different colors on the map mean. How will participants know what would be a good path to draw? For future work, perhaps an alternate approach could have the user navigate a virtual entity through an environment themselves to show what path they would take.

---

### Official Review · Reviewer_Qt9G · 2024-02-24
**Accept**

**Rating:** 9
**Confidence:** 5

**Review:**

The paper outlines a method to improve robot predictability in human-robot collaboration using trajectory optimization. The method involves defining a trajectory as an ordered list of waypoints. The optimization process is stochastic and considers various objective functions such as collision avoidance, valid states, connectivity, distance from the optimal trajectory, and symmetry to balance predictability and optimality. The authors suggest using virtual reality to test this approach without putting people at risk while they implement the method on a Boston Dynamics Spot robot. They plan to carry out two tests with users, one using virtual reality for a realistic yet safe experience, and another one online.


Strengths:

1. Innovative approach of enhancing predictability of robot movements using trajectory optimization.
2. Integration of virtual reality for safe and diverse testing environments.
3. Clearly defined hypothesis and experiment design for experimental validation.


Improvement potential:

1. Limited real-world testing; the use of VR might not fully capture the complexities of physical interactions.
2. Due to a lack of experimental implementations, it is unclear about the practical usability of the trajectory algorithm while involving people in the suggested situation.
3. The focus on predictability might compromise the efficiency or optimality of robot movements in certain scenarios. Lacking a mechanism to prioritize one over the other.
4. References to figures were insufficient in the paper leading to problems in correlating the text with the visuals provided. Specifically, Figure 2 would be more informative if it included detailed descriptions of the various color zones to facilitate a better understanding of the environment depicted. In Figure 3, the path presumably represents segments as mentioned in the Methods section, however, a more explicit explanation within the figure could enhance comprehension. Additionally, Figure 4 fails to inform about the task to be performed for the online study.

In summary, I think this paper is a good fit for VAM-HRI, and I recommend acceptance.

---

### Decision · Program_Chairs · 2024-02-26

Accept (Oral)